# Association of Dental Fluorosis and Urinary Fluoride with Intelligence among Schoolchildren

**DOI:** 10.3390/children10060987

**Published:** 2023-05-31

**Authors:** Yuh-Yih Lin, Wen-Yu Hsu, Chin-En Yen, Suh-Woan Hu

**Affiliations:** 1Institute of Oral Sciences, Chung Shan Medical University, Taichung 40201, Taiwan; yuhyih@csmu.edu.tw; 2Department of Psychology, Chung Shan Medical University, Taichung 40201, Taiwan; wyhsu@csmu.edu.tw; 3Department of Early Childhood Development and Education, Chaoyang University of Technology, Taichung 41349, Taiwan; ceyen@cyut.edu.tw; 4Department of Dentistry, Chung Shan Medical University Hospital, Taichung 40201, Taiwan

**Keywords:** children, dental fluorosis, fluoride, intelligence, urinary fluoride

## Abstract

Fluoride is present naturally in water and has been used worldwide for the prevention of caries. Several studies conducted in high water fluoride or endemic fluorosis areas reported that fluoride adversely affected children’s cognitive function, but some studies had negative findings. This study aimed to assess the relationship between urinary fluoride, dental fluorosis, and intelligence among schoolchildren living in communities with non-fluoridated drinking water. This cross-sectional study was conducted on 562 children aged 6–12 years in Taichung, Taiwan. Each child’s urinary fluoride level was determined by a fluoride-ion-selective electrode, and the dental fluorosis condition was evaluated according to the criteria of Dean’s Index. The Raven’s Colored Progressive Matrices-Parallel and Standard Progressive Matrices-Parallel were used to assess children’s intelligence. The results showed that the mean (±standard deviation) urinary fluoride concentrations were 0.40 ± 0.27 mg/L (0.43 ± 0.23 mg/g creatinine) among participants. The prevalence of dental fluorosis was 23.67%. After extensive evaluation of potential confounders, dental fluorosis and urinary fluoride were not associated with intelligence quotient (IQ) scores or grades in the regression models. In conclusion, dental fluorosis and urinary fluoride levels were not significantly related to the IQ of schoolchildren living in areas with low drinking water fluoride.

## 1. Introduction

Fluoride is an abundant element in the earth’s crust and occurs naturally in water [1]. In addition, fluoride has been used worldwide as a public health measure, such as community water fluoridation, salt fluoridation, fluoride supplements, fluoride varnishes, and fluoride toothpaste for the prevention of dental caries [2]. Children are exposed to fluoride mainly through the consumption of drinking water and the ingestion of foods, beverages, and dental products [1].

There have been growing concerns about adverse health effects from excessive fluoride exposure, especially the potential impact on children’s cognitive function [3,4,5]. In the past 20 years, many studies have investigated the association of urinary fluoride levels and dental fluorosis with intelligence quotient (IQ) scores among school-age children, and the results were inconsistent. While the majority of the studies reported a significant reverse relationship between urinary fluoride concentrations and IQ [6,7,8,9,10,11,12,13,14,15,16,17,18,19], three reviewed studies did not observe a significant association [20,21,22]. Furthermore, several studies reported that dental fluorosis was significantly associated with lower IQ among school-age children [8,12,21,23,24,25,26,27]. However, some did not find a significant relationship [9,16,22].

The discrepancies in previous findings could be related to differences in study participants’ fluoride exposure levels, in the evaluation of potential confounding factors, or in consideration of other sources of fluoride exposure. Furthermore, most of these studies were conducted among children living in areas with naturally higher water fluoride levels and/or in endemic fluorosis areas in China, India, Iran, Indonesia, Mexico, and Pakistan. The relationship of urinary fluoride and dental fluorosis with IQ among children living in other territories, especially in regions with non-fluoridated public drinking water, is not well understood and merits clarification.

The public water supply systems in Taiwan, which serve 95% of the population, are not fluoridated and have a maximum limit of 0.8 mg/L for fluoride [28,29]. To prevent dental caries, Taiwan started the salt fluoridation program in 2016. Fluoride can be added to small packages (≤1 kg) of table salt with a maximum concentration of 200 mg/kg, and consumers are free to purchase salt with or without fluoride added [30]. However, the fluoride exposure levels among Taiwanese children and whether the exposures influence children’s intelligence remains unclear and needs further investigation. In this study, both urinary fluoride concentrations and dental fluorosis were measured to estimate an individual’s fluoride exposure. The purpose was to assess the association of urinary fluoride levels and dental fluorosis with IQ among schoolchildren living in areas without water fluoridation, with an extensive evaluation of confounding factors, dietary fluoride intake, and usage of dental fluoride products.

## 2. Materials and Methods

### 2.1. Study Design and Participants

This cross-sectional study was conducted on schoolchildren in Taichung, Taiwan, between October 2019 and May 2020. Details of the study design and sampling methods were described elsewhere [31]. Briefly, five elementary schools, which were randomly sampled from districts with different urbanization levels, participated in this study. In each school, we invited students from two classes from each grade to participate. Invitation letters were sent to 1380 students from 60 classes, and 562 children who completed the measurement of urinary fluoride, the evaluation of dental fluorosis, and the assessment of intelligence were included in the data analysis. Written informed consent was obtained prior to each child’s participation in the research. Approval for this study was granted by the Institutional Review Board of the Chung Shan Medical University Hospital (CSMUH No. CS18238) in Taichung, Taiwan.

### 2.2. Evaluation of Dental Fluorosis

Two trained and calibrated dentists conducted oral examinations, including evaluation of dental fluorosis, for all the participants following the guidelines of the World Health Organization’s Oral Health Surveys-Basic Methods [32]. Each child was assessed for dental fluorosis according to the criteria of Dean’s Index [32]. The Index has the following categories: normal (smooth, glossy, and creamy-white enamel surfaces), questionable (slight deviation in enamel translucency), very mild (scattered small, opaque, paper-white areas, <25% of tooth surface), mild (white opacities of enamel surface, 25–50%), moderate (apparent wear and brown staining of enamel surfaces), and severe (marked enamel hypoplasia with widespread pitted areas and brown stains) [32].

### 2.3. Measurement of Urinary Fluoride and Creatinine

One spot urine sample of about 20 mL was collected directly using a sterile polyethylene wide-mouth bottle from each child. The collected samples were placed in ice immediately and brought to the laboratory. Then, the samples were transferred to polypropylene tubes and stored in freezers at −20 °C until analysis. A fluoride-ion-selective electrode (perfectION^TM^ combination fluoride electrode, Mettler Toledo, Greifensee, Switzerland) was used to determine fluoride concentration for these urine samples. In brief, fluoride standard solutions of 0.2, 0.5, 1.0, 2.0, and 5.0 mg/L were prepared by diluting a 1000 mg/L fluoride standard solution in distilled water. To measure the fluoride concentration, the urine samples and standards were allowed to stand at room temperature, and TISAB II solution (Total Ionic Strength Adjustment Buffer II) was then added to the test tubes at a ratio of 1:1 and shaken until stabilized. The direct readings of fluoride concentration (expressed in mg/L) were obtained by following the manufacturer’s instructions.

The concentration of creatinine in each urine sample was measured by Jaffe reaction on a Hitachi 7180 Automatic Analyzer (Hitachi High-Tech Corp., Tokyo, Japan). Urinary creatinine was used to adjust the measured fluoride concentrations in the urine samples [33,34]. The creatinine-adjusted levels of fluoride were expressed in mg/g creatinine.

### 2.4. Assessment of Intelligence

The Taiwan edition of Raven’s Colored Progressive Matrices-Parallel, Standard Progressive Matrices-Parallel, and the test manual compiled by Chen and Chen [35,36,37] were used to assess the intelligence of children in first and second grades and those in third to sixth grades, respectively. These Raven tests are well-validated and broadly applied methods for evaluating children’s basic cognitive ability [35,38]. In this study, all tests were administered in the children’s classrooms by the same qualified psychologist following the guidance of the test manual [35]. The original scores from the tests were converted into standardized IQ scores (mean = 100, standard deviation = 15) as well as percentile ranks according to the age norms of children in Taiwan [35]. Subsequently, the children’s percentile ranks were categorized, according to the classification of the test manual, into the following five IQ grade groups: A—intellectually superior (≥95th percentile); B—definitely above the average in intellectual capacity (75–94th percentiles); C—intellectually average (25–74th percentiles); D—definitely below average in intellectual capacity (6–24th percentiles); and E—intellectually impaired (≤5th percentile) [35]. For the data analysis, these five IQ grades were further combined into three groups: intellectually above average (Grades A and B), intellectually average (Grade C), and intellectually below average (Grades D and E) or into two groups: intellectually average or above (Grades A, B and C) vs. below average (Grades D and E).

### 2.5. Questionnaire Survey and Anthropometric Measurement

A structured questionnaire was used to collect information on participating children’s age, gender, history of physician-diagnosed diseases, food intake habits (e.g., the frequencies for intake of foods with high fluoride), usage of systemic and topical fluoride products, and parents’ educational levels. Each child’s parents or guardians answered the questionnaire. The child’s body weight and height were measured by one trained researcher. The formula for calculating body mass index (BMI) was weight (kg)/height (m)^2^.

### 2.6. Statistical Analysis

The descriptive statistics for all important variables were checked first. The box plots and the Kolmogorov–Smirnov test were used to evaluate the normality of distributions for continuous variables. Then, the bivariate analyses were conducted to assess the un-adjusted relationship between IQ score, dental fluorosis, urinary fluoride, and other factors. The Student’s *t*-test, analysis of variance (ANOVA) followed by Tukey’s post hoc test, and Pearson correlation coefficients were applied to compare means of normally distributed variables, such as IQ scores, between and among groups or to evaluate the correlations between these variables. The chi-squared test was used to compare percentages of categorical variables. Since urinary fluoride variables were not normally distributed, the non-parametric methods, including the Wilcoxon rank-sum test, Kruskal–Wallis test, and Spearman rank correlation, were applied to compare urinary fluoride levels between/among groups or to estimate their correlations with other continuous variables. For further multivariable analysis, urinary fluoride variables were transformed by natural logarithm (ln) transformation. Finally, multiple regression analyses were performed to assess the adjusted association between IQ scores and dental fluorosis as well as between IQ scores and ln-transformed urinary fluoride, adjusting for age, sex, and potential confounding factors identified in the bivariate analyses. The multivariable logistic regression was used to assess the relationship between IQ grade (below average vs. intellectually average or above) and dental fluorosis or urinary fluoride, respectively, adjusting for important factors. Moreover, the multiple regression analysis was implemented to evaluate the relationship between ln-transformed urinary fluoride and dental fluorosis, with adjustment for potential confounders. The SAS version 9.4 software (SAS Institute Inc., Cary, NC, USA) was utilized for all data analyses. A significance level of 0.05 was used for the statistical tests.

## 3. Results

### 3.1. Descriptive Characteristics, Dental Fluorosis, Urinary Fluoride, and IQ

Table 1 shows the descriptive characteristics, childhood illness, usage of systemic and topical fluoride, urinary fluoride levels, and IQ scores for all participating children as well as stratified by their dental fluorosis status. In total, 133 children had questionable (*n* = 95), very mild (*n* = 33) or mild (*n* = 5) dental fluorosis, with an overall prevalence of 23.67%. None of the children had moderate or severe dental fluorosis. The age of participants ranged from 6.33 to 12.71 years. Allergic rhinitis was significantly more prevalent among children with dental fluorosis. Other physician-diagnosed diseases were reported by less than 5% of the participants and, therefore, were not included in the analysis. Intake of systemic dental fluoride products was reported by 15.48% of the participants. The usage of systemic and topical fluoride products was not associated with dental fluorosis status. The frequency of seafood consumption was significantly related to dental fluorosis status.

The mean IQ scores or the frequencies of IQ grades were not significantly different between children with and without dental fluorosis. Other factors listed in Table 1 and significantly associated with IQ scores were the father’s and mother’s education levels (ANOVA, both *p*-values < 0.01), ever receiving professional fluoride varnish treatment (*t*-test, *p*-value < 0.01), and using fluoride toothpaste daily (*t*-test, *p*-value < 0.05). Children with higher parental education levels had significantly higher mean IQ scores (college and above > high school or less than high school, Tukey’s post hoc test, *p*-value < 0.05).

Urinary fluoride ranged from 0.03 to 2.08 mg/L, with a geometric mean of 0.32 mg/L. The creatinine-adjusted urinary fluoride ranged from 0.08–2.27 mg/g creatinine, with a geometric mean of 0.39 mg/g creatinine. Urinary fluoride levels, with or without adjusting for urinary creatinine, were significantly higher in children with dental fluorosis than in those without fluorosis. Tea consumption was significantly associated with urinary fluoride, with or without creatinine adjustment (Kruskal–Wallis test, both *p*-values < 0.001), and children who drank tea more frequently had higher urinary fluoride levels. Usage of systemic and topical fluoride products, IQ grade, and other factors listed in Table 1 were not significantly associated with urinary fluoride concentrations.

### 3.2. Correlation of Urinary Fluoride and IQ Scores

The urinary fluoride levels, with or without creatinine adjustment, were not significantly correlated with IQ scores (Table 2). Urinary fluoride (mg/L) was significantly correlated with creatinine-adjusted urinary fluoride (mg/g creatinine). Age and BMI had significant positive correlations with urinary fluoride (mg/L) and significant negative correlations with creatinine-adjusted urinary fluoride (mg/g creatinine).

### 3.3. Adjusted Relationship between Urinary Fluoride and Dental Fluorosis

Results from the multiple linear regression analyses indicated that children with dental fluorosis had significantly higher creatinine-adjusted urinary fluoride levels than those without dental fluorosis (ratio of the geometric means between two groups = 1.14, 95% confidence interval: 1.04–1.25) after controlling for age, gender, body mass index, area of school, father’s education level, mother’s education level, history of allergic rhinitis, and tea consumption in the regression models (Table 3). The result was marginally significant for urinary fluoride without creatinine adjustment (ratio = 1.14, 95% confidence interval: 1.00–1.30). Further inclusion of variables for fluoride usage in the regression models did not alter the results.

### 3.4. Adjusted Association between IQ and Dental Fluorosis or Urinary Fluoride

Dental fluorosis and urinary fluoride were not significantly related to IQ scores after adjusting for age, gender, body mass index, area of school, father’s education level, mother’s education level, and history of allergic rhinitis (Table 4). The father’s education level and area of school were significantly associated with IQ scores in the multiple linear regression models. Furthermore, results of the multivariable logistic regression analyses showed that neither dental fluorosis nor urinary fluoride was significantly associated with IQ grade (below average vs. intellectually average or above average), with adjustment for important factors.

## 4. Discussion

In this study, both dental fluorosis and urinary fluoride were not significantly associated with IQ among 6–12-year-old schoolchildren living in areas without water fluoridation after adjusting for age, sex, BMI, father’s education, mother’s education, area of school, and history of allergic rhinitis. The negative finding for dental fluorosis and IQ was consistent with three previous studies [9,16,22] but was inconsistent with several studies of school-age children [8,12,21,23,24,25,26,27]. The non-significant association between urinary fluoride and IQ was consistent with three previous studies [20,21,22] but was conflicting with many other studies [6,7,8,9,10,11,12,13,14,15,16,17,18,19].

There are several possible interpretations for the negative findings in the present study. First, the prevalence and severity of dental fluorosis observed in this study were lower compared to previous studies, which reported a significant association between dental fluorosis and IQ among schoolchildren [8,12,21,24,25,26,27]. These studies reported a dental fluorosis prevalence (and severity) of 21% (mild/moderate/severe) [24] to 68% (mild/moderate/severe) [8] or 91.6% (mild/moderate) [12]. Furthermore, the studies by Ding et al. [9], Saeed et al. [16], and Soto-Barreras et al. [22] did not find a significant IQ-dental fluorosis relationship, but the prevalence and severity of dental fluorosis, ranging from 42.6% (very mild~moderate) [9] to 52.2% (mild~severe) [22] and 67.8% (mild~severe, exposed group) [16], were much higher than that in the current study.

Second, children participating in the present study had lower urinary fluoride levels than children in previous studies that found a significant IQ-urinary fluoride relationship [6,7,8,9,10,11,12,13,14,15,16,17,18,19]. For example, these studies reported mean (± SD) urinary fluoride values of 0.41 ± 0.49 mg/L [12] to 7.01 ± 1.02 mg/L [6]. In the studies by Bashash et al. [20], Choi et al. [21], and Soto-Barreras et al. [22], which reported a negative association between IQ and urinary fluoride, the concentrations were also higher than those observed in the present study. The mean (±SD) or the geometric mean was 0.82 ± 0.38 mg/L with specific gravity adjustment [20], 1.64 mg/L [21], and 0.48 ± 0.23~0.67 ± 0.41 mg/L (for normal~severe fluorosis groups) [22]. Moreover, in the 2014-2015 Canadian national survey, 6–11-year-old children had a geometric mean urinary fluoride of 0.47 mg/L or 0.50 mg/g creatinine [39], which was higher than the levels in the present study. Note that all nine water treatment plants in Taichung had mean drinking water fluoride levels ranging from 0.02 to 0.14 mg/L [40]. In addition, usage of systemic fluoride products, such as fluoride tablets and fluoridated salt, was rarely reported by participating children (Table 1). Children in this study appeared to have low fluoride intakes from drinking water and dental products and, therefore, low urinary fluoride levels.

Third, current urinary fluoride might not represent long-term fluoride exposures, and measurement errors in the estimation of exposure levels might have biased the results. To further evaluate the relationship of urine fluoride levels at different time periods, we collected a second urine sample, three months after the first sample collection, from 316 children participating in the present study. Urinary fluoride levels, with and without creatinine adjustment, from these two samples, were significantly correlated, with a Spearman rank correlation coefficient of 0.50 and 0.42, respectively). Nonetheless, these children’s current urinary fluoride levels might not represent the exposure that occurred in their early childhood or in the prenatal period when the developing nervous system was vulnerable to environmental toxicants [41]. Several prospective birth cohort studies conducted in Canada and Mexico reported that infants or children with higher maternal fluoride exposure had significantly lower cognitive function [20,42,43,44,45]. Furthermore, the study by Bashash et al. [20] found a significant reverse association between maternal creatinine-adjusted urinary fluoride and 6–12-year-old children’s IQ scores but a non-significant cross-sectional relationship between these children’s urinary fluoride levels and IQ scores. These findings may imply that the prenatal period is a more critical time for the potential neurotoxic effects of fluoride.

Fourth, the potential misclassification of dental fluorosis status could have biased the study results. Dental fluorosis is a sensitive effect biomarker for excessive fluoride intake during critical periods of tooth development, except the third molars, up to 7–8 years of age [46,47]. However, 78% of the children participating in this study had mixed dentition, and 21% had permanent teeth only, according to the results of the oral examination. In this cross-sectional study, we were not able to evaluate whether the children’s exfoliated primary teeth or unerupted permanent teeth were affected by fluorosis and, therefore, might have misclassified the participants’ dental fluorosis status.

Fifth, although several important factors were adjusted in the regression models, the study results could be biased toward the null by other confounding variables. Several factors considered in previous studies of the IQ–fluoride relationship, such as birthweight or low birthweight [7,17], abnormal birth, and maternal age at delivery [14], were not evaluated in this study. In this study, the father’s education level was significantly associated with children’s IQ scores in the regression models, which was consistent with previous findings [48]. Lastly, the findings from the current study suggest that the fluoride exposure levels among children residing in non-fluoridated communities are low and have no obvious effects on children’s IQ. Longitudinal studies in New Zealand and Australia, respectively, reported that exposure to fluoridated tap water (with 0.7–1.0 ppm and 0.6–1.1 ppm fluoride) by age five years was not significantly associated with IQ at age 7–13 and 38 years [49] or behavioral development and executive functioning during adolescence [50].

In the present study, we did not observe a significant association between IQ and dental fluorosis or urinary fluoride among 6–12-year-old children living in areas without water fluoridation after extensive evaluation of confounding factors. Both urinary fluoride concentrations, an exposure biomarker, and dental fluorosis, an effect biomarker for excessive fluoride ingestion before age 7–8 years, were applied to estimate participating children’s fluoride exposures. Fluoride intake from the usage of dental products and food consumption were considered. In addition, urinary creatinine was used to adjust for variation in the urinary dilution of spot urine samples [33,34]. Moreover, this study had a larger sample size and higher statistical power compared to previous studies with negative findings for IQ and dental fluorosis [9,16,22] or urinary fluoride [20,21,22]. Results from this study help to clarify the relationship of low-level fluoride exposures and children’s cognitive function. However, this study has several limitations. First, only the Raven’s Progressive Matrices test was used to evaluate each child’s basic cognitive function. This test measures fluid intelligence but not crystal intelligence and other cognitive abilities [51]. The negative finding in this study did not necessarily exclude the potential effects of fluoride exposure on all aspects of cognitive functions in schoolchildren. Most previous studies that reported a significant IQ–urinary fluoride or IQ–dental fluorosis relationship applied Raven’s test to evaluate school-age children’s intelligence [6,7,8,9,10,11,12,13,14,17,18,19,23,24,25,26,27], and some used the Wechsler Intelligence Scale [15,16,21]. Future studies may apply various tests which measure different intelligence constructs to have a more comprehensive assessment of the children’s cognitive abilities. Second, it was difficult to clarify the temporal relationship between the urinary fluoride measured and the current IQ scores in this cross-sectional study. Although the same study design was used in studies that reported a significant IQ–urinary fluoride association among schoolchildren [6,7,8,9,10,11,12,13,14,15,16,17,18,19], future investigations with a cohort study design may help to better understand the effects of early childhood fluoride exposure on intelligence. Third, about 41% of the schoolchildren who were invited to participate in this study completed the evaluation of dental fluorosis, the IQ test, and the measurement of urinary fluoride. Because the COVID-19 pandemic started during the study period, the data collection schedule was interrupted, and the recruitment of participants was affected. Finally, although many potential confounding factors were evaluated in the present study, several factors, as mentioned above, were not assessed and might have confounded the association.

In conclusion, dental fluorosis and urinary fluoride levels were not significantly associated with IQ scores among 6–12-year-old schoolchildren living in areas with low drinking water fluoride concentrations after adjusting for confounding factors. The prevalence and severity of dental fluorosis and urinary fluoride levels were low among these children. Future studies with long-term follow-up and a more comprehensive assessment of intelligence will better elucidate the potential effects of low fluoride exposures on children’s cognitive capacities.

## Figures and Tables

**Table 1 children-10-00987-t001:** Characteristics, urinary fluoride, and IQ scores of participating children according to dental fluorosis status (*n* = 562).

		Dental Fluorosis		
Variable	All Children	Yes	No	*p*
*n*	562	133	429	
Age (years)	9.51 ± 1.76	9.79 ± 1.47	9.43 ± 1.84	0.021 ^a^
Gender				
Boys	49.82	50.38	49.65	0.884 ^b^
Girls	50.18	49.62	50.35	
Area of school				
Urban	21.71	10.53	25.17	0.001 ^b^
Sub-urban	37.72	40.60	36.83	
Rural	40.57	48.87	38.00	
Father’s education level				
Less than high school	11.21	10.53	11.42	0.861 ^b^
High school	40.04	42.11	39.39	
College and above	47.69	46.62	48.02	
Mother’s education level				
Less than high school	9.07	6.77	9.79	0.533 ^b^
High school	33.63	34.59	33.33	
College and above	55.16	57.89	54.31	
Body mass index (kg/m^2^)	18.38 ± 3.72	18.61 ± 3.97	18.31 ± 3.63	0.413 ^a^
History of allergic rhinitis				
Yes	20.64	27.82	18.41	0.017 ^b^
No	77.76	69.92	80.19	
Seafood consumption				
<1 time/month	12.28	18.80	10.26	0.049 ^b^
1–3 times/month	24.73	19.55	26.34	
1–6 times/week	43.59	43.61	43.59	
≥1 time/day	18.51	18.05	18.65	
Tea consumption (black tea, oolong tea or green tea)				
<1 time/month	40.21	42.86	39.76	0.862 ^b^
1–3 times/month	30.96	30.08	31.53	
1–6 times/week	21.35	19.55	22.12	
≥1 time/day	6.76	7.52	6.59	
Usages of systemic fluoride				
Fluoride tablets				
Ever used	12.81	13.53	12.59	0.789 ^b^
Never used/do not remember	86.83	86.47	86.95	
Fluoridated salt				
Ever used	4.27	3.76	4.43	0.713 ^b^
Never used/do not remember	94.48	96.24	93.94	
Usages of topical fluoride				
Fluoride varnish				
Ever used	91.10	93.98	90.21	0.284 ^b^
Never used/do not remember	8.19	6.02	8.86	
Use fluoride toothpaste daily				
Yes	71.53	70.68	71.79	0.633 ^b^
No	26.69	28.57	26.11	
Participated in school fluoride mouthrinse program				
Yes	79.00	75.94	79.95	0.208 ^b^
No	18.68	22.56	17.48	
Urinary fluoride levels				
Urinary fluoride (mg/L)	0.40 ± 0.27	0.46 ± 0.32	0.38 ± 0.25	
Median (Q1, Q3)	0.33 (0.21, 0.52)	0.37 (0.26, 0.55)	0.32 (0.20, 0.51)	0.009 ^c^
Creatinine adjusted urinary fluoride (mg/g creatinine)	0.43 ± 0.23	0.48 ± 0.27	0.42 ± 0.22	
Median (Q1, Q3)	0.38 (0.29, 0.51)	0.41 (0.33, 0.56)	0.38 (0.28, 0.49)	0.002 ^c^
IQ scores and grades				
IQ scores	98.08 ± 15.28	99.22 ± 15.87	97.72 ± 15.09	0.325 ^a^
IQ grades				
Above average	25.44	28.57	24.48	0.600 ^b^
Intellectually average	47.51	46.62	47.79	
Below average	27.05	24.81	27.74	

Values are given as column % or arithmetic mean ± standard deviation, unless otherwise specified. Q1: 25th percentile; Q3: 75th percentile. ^a^ 2-sample *t*-test for comparing means of continuous variables. ^b^ Chi-squared test for comparing percentages of categorical variables. ^c^ Wilcoxon rank sum test for comparisons of non-normally distributed variables.

**Table 2 children-10-00987-t002:** Correlations between urinary fluoride levels, IQ scores, and other factors.

Variables	IQ Score	Urinary Fluoride	Creatinine Adjusted Urinary Fluoride
Age	0.03 ^a^	0.12 ^b^ ***	−0.11 ^b^ **
Body mass index	−0.01 ^a^	0.10 ^b^ *	−0.10 ^b^ *
IQ score	1	−0.04 ^b^	−0.05 ^b^
Urinary fluoride		1	0.52 ^b^ ***
Creatinine adjusted urinary fluoride			1

^a^ Pearson correlation coefficient, *n* = 562. ^b^ Spearman rank correlation coefficient, *n* = 562. * *p* < 0.05, ** *p* < 0.01, *** *p* < 0.001.

**Table 3 children-10-00987-t003:** Relationship between urinary fluoride and dental fluorosis results from the multiple linear regression analyses.

Dependent Variable	Independent Variable	Adjusted R^2^	Parameter Estimate ^b^ (95% CI)	Ratio (95% CI) ^c^ for the Geometric Mean of Urinary Fluoride
Model 1		0.07		
Urinary fluoride ^a^	Dental fluorosis			
	No		0	1
	Yes		0.13 (−0.00–0.26)	1.14 (1.00–1.30)
			*p* = 0.055	
Model 2		0.07		
Creatinine adjusted urinary fluoride ^a^	Dental fluorosis			
	No		0	1
	Yes		0.13 (0.04–0.22)	1.14 (1.04–1.25)
			*p* = 0.005	

CI: confidence interval. ^a^ With natural logarithm transformation. ^b^ Adjustment for age, gender, body mass index, area of school, father’s education level, mother’s education level, history of allergic rhinitis, and tea consumption. ^c^ Exponential of parameter estimate and its 95% CI.

**Table 4 children-10-00987-t004:** Associations between IQ scores, urinary fluoride, and dental fluorosis, results from the multiple linear regression analyses.

Dependent Variable	Independent Variable	Adjusted R^2^	Parameter Estimate ^b^ (95% CI)	Change (95% CI) in Mean IQ Score for a 10% Increase in Urinary Fluoride
Model 1		0.06		
IQ scores	Dental fluorosis			
	No		0	
	Yes		0.94 (−2.06–3.93)	
			*p* = 0.540	
Model 2		0.06		
IQ scores	Urinary fluoride ^a^		−0.97 (−2.90–0.97)	−0.092 (−0.28–0.09)
			*p* = 0.327	
Model 3		0.07		
IQ scores	Creatinine adjusted urinary fluoride ^a^		−2.21 (−4.99–0.57) *p* = 0.119	−0.21 (−0.48–0.05)

CI: confidence interval. ^a^ With natural logarithm transformation. ^b^ Adjustment for age, gender, body mass index, area of school, father’s education level, mother’s education level, and history of allergic rhinitis.

## Data Availability

Data are available upon reasonable request.

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
