# Peer review of "Association of Dental Fluorosis and Urinary Fluoride with Intelligence among Schoolchildren"

_children, 2023, doi:10.3390/children10060987_

Round 1

Reviewer 1 Report

I thought the study was well designed and presented 

The discussion is very good and the point made that the fluoride levels at ages 6 to 12 may not reflect the levels during intrauterine and early development

Consideration might be given to noting that the bashash et al study which found an association between iq  and fluoride exposure during pregnancy did not find any association at age 6

Reviewer 2 Report

The manuscript made by Lin YY et al., is interesting and well redacted. The authors made a great effort to measure several variables related to fluorosis and IQ. Despite the interesting nature of this study, I have some questions and suggestions that the authors need to resolve.

First, the title of the manuscript is not completely related to the study; I suggest including in it the low ingest of fluorosis.

Second, in the paragraph about "the higher level of fluorosis in several countries," I suggest including references (9, 16, 22) and articles related to low exposure to fluorosis.

Third. In the discussion, the authors included two variables, polymorphism and arsenic, that were not included in their study. I suggest not discussing both variables (polymorphism and fluorosis).

Fourth. Please include the most representative photographs of dental fluorosis.

Fifth. In the discussion in the paragraph that says, "The negative finding..." would it be possible to make a better redaction and give some results of the manuscript evaluated, for example: "Bashas et al., describing that relationship between IQ and fluorosis was not completely related, same in our study we found..."

Sixth. "There are several possible..." In this paragraph, I suggest that authors make some hypotheses about their results by comparing them with those of other authors, and it is possible to compare their results with those of articles with mild fluorosis.

Overall comments:

The manuscript is interesting; the authors made a great effort to evaluate the association between fluorosis and IQ in areas with a low grade of fluorosis. Despite the design of the methodology and the interpretation of the results being interesting, the discussion feels weak, Authors need to strengthen their discussion with hypotheses from their results and compare their results better with those of other authors. Maybe find articles with the same results to make a more proper discussion. The sociodemographic variables (age, gender, school, education level, etc.) were not widely discussed; please review.

English writing is correct, but despite the correct language, I suggest it be reviewed with an expert to improve the redaction and punctuation.

Reviewer 3 Report

FLUOROSIS ARTICLE REPORT.  

 The entire article should be submitted according to the journal's guidelines.  

 The title should be revised. It is not clear.  

 In the Introduction there are annotations that seem to correspond to a version that should not be the final version of the paper (it refers to pages.).  

The association is not between fluoride in urine and QI, but should be between the potential effects of an excess of fluoride supplied to the organism (which is evidenced by a high concentration in excreted urine) and QI.  

 The Material and Method section is given under a different name and should be corrected.  

 The procedure is very well described although the tables should not be so large. In particular, Table 1 should be much more summarized, highlighting only the important information and with the format recommended by the publisher.  

 Tables 2 and 3 are correct and very clarifying.  

 The discussion is correct as well as the bibliography.

Round 2

Reviewer 2 Report

The authors responded to all my questions and suggestions; the manuscript was improved. I do not have any other questions or suggestions.

English is adequately redacted; however, I suggest the manuscript be reviewed by an expert for better punctuation and redaction.

Reviewer 3 Report

The article has improved since the first version